# Absent in Melanoma 2 Mediates Inflammasome Signaling Activation against *Clostridium perfringens* Infection

**DOI:** 10.3390/ijms25126571

**Published:** 2024-06-14

**Authors:** Zhaoguo Ma, Yanan Lou, Na Wang, Yi Zhao, Shuxin Zhang, Mingyue Zhang, Jiaqi Li, Qian Xu, Aobo He, Shuixing Yu

**Affiliations:** State Key Laboratory of Reproductive Regulation and Breeding of Grassland Livestock, College of Life Sciences, Inner Mongolia University, Hohhot 010070, China; 15935808653@163.com (Z.M.); 15847200038@163.com (Y.L.); 15047616541@163.com (N.W.); 13804752339@163.com (Y.Z.); zsx1230205@163.com (S.Z.); zhangmy0622@163.com (M.Z.); 15847769721@163.com (J.L.); xq15174941516@163.com (Q.X.); 13104376383@163.com (A.H.)

**Keywords:** *C*. *perfringens*, gas gangrene, AIM2, inflammasome, innate immune

## Abstract

Absent in melanoma 2 (AIM2), a key component of the IFI20X/IFI16 (PYHIN) protein family, is characterized as a DNA sensor to detect cytosolic bacteria and DNA viruses. However, little is known about its immunological role during pathogenic *Clostridium perfringens* (*C*. *perfringens*) infection, an extracellular bacterial pathogen. In a pathogenic *C*. *perfringens* gas gangrene model, *Aim2*^−/−^ mice are more susceptible to pathogenic *C*. *perfringens* soft tissue infection, revealing the importance of AIM2 in host protection. Notably, *Aim2* deficiency leads to a defect in bacterial killing and clearance. Our in vivo and in vitro findings further establish that inflammasome signaling is impaired in the absence of *Aim2* in response to pathogenic *C*. *perfringens*. Mechanistically, inflammasome signaling downstream of active AIM2 promotes pathogen control. Importantly, pathogenic *C*. *perfringens*-derived genomic DNA triggers inflammasome signaling activation in an AIM2-dependent manner. Thus, these observations uncover a central role for AIM2 in host defense and triggering innate immunity to combat pathogenic *C*. *perfringens* infections.

## 1. Introduction

*C. perfringens*, one of the most common gas gangrene-associated anaerobes in humans, is ubiquitous in different environments, including soil, water, food, and even the gastrointestinal tracts of mammals, and it can also cause enterocolitis [1,2,3]. *C. perfringens* gas gangrene can develop into a traumatic tissue infection in as little as 6–8 h, with shock, myonecrosis, and organ exhaustion in 50% of patients, among these 40% die. Even with many of the advances of modern medicine, radical amputation in an emergency is still the single best treatment [4]. Meanwhile, the Centers for Disease Control and Prevention predicts *C. perfringens* cause almost 1 million foodborne illnesses in the USA annually [5]. Worse, clostridial necrotizing enteritis can cost $ 2 billion to $ 6 billion each year in the global livestock industry [6]. Despite growing scientific concern over antibiotic resistance, the recommended treatment for *C. perfringens* infections remains the combination of penicillin with clindamycin or carbapenem [7]. So, it is necessary to develop effective preventive and therapeutic strategies for control of pathogenic *C. perfringens* infections.

Innate immunity serves as the first line of defense in perceiving and eliminating invasive pathogens [8]. Microbe-associated molecular patterns or endogenous danger-associated molecular patterns can be sensed by pattern recognition receptors (PRRs) to prime innate immunity. PRRs that have been characterized include cytosolic DNA sensors, RIG-I-like receptors, NOD-like receptors, Toll-like receptors, and C-type lectin receptors [9]. Previously, we have provided evidence that MLKL-NLRP3-extracellular traps and GPR120-NLRP3 signaling axes protect against *C. perfringens* infection by enhancing bactericidal activity [10,11]. Also, Toll-like receptor 4 signaling contributes to host defense against *C. perfringens* infection through replenishment of neutrophils and elimination of bacteria [12]. Therefore, research on the host innate immune defense mechanism will be conducive to implementing combat strategies for pathogenic *C. perfringens* infections.

Using subtractive cDNA selection, absent in melanoma 2 (AIM2) was identified in the human malignant melanoma cell line UACC-903(+6) [13]. It was initially considered to be a novel member of the interferon-induced PYHIN protein, including four members in humans (e.g., AIM2, IFI16, IFIX, and MNDA) and homologues in mice (e.g., *Aim2*/p210, p202, p204, and p205) [14]. Subsequently, it was found that AIM2 is composed of C-terminal HIN-200 and N-terminal PYD structural domains, which remain inactive during homeostasis [15]. Nowadays, exploring AIM2 has primarily focused on its role in pathogenic infections through inflammasome-dependent or non-inflammasome-dependent signals. Evidence has shown that the AIM2 inflammasome signaling is involved in *Streptococcus pneumoniae*, *Listeria monocytogenes*, *Porphyromonas gingivalis*, and *Aspergillus fumigatus* infections [16,17,18,19]. Additionally, AIM2 is essential for the maintenance of intestinal integrity by Akt during foodborne *Salmonella* infection [20]. However, little is known about the biological role of AIM2 in pathogenic *C. perfringens* infection.

In the current study, we sought to define the immunological role of AIM2 in response to pathogenic *C*. *perfringens* invasion, and reveal it might elicit host antipathogenic innate immunity. Our findings show that AIM2 is required for host defense against pathogenic *C*. *perfringens* gas gangrene. *Aim2*^−/−^ mice are more susceptible to pathogenic *C*. *perfringens* infection, with lower survival rates, increased bacterial colonization, disrupted muscle architecture, and severe tissue injuries. Obviously, *Aim2* deficiency results in impaired inflammasome signaling activation following pathogenic *C*. *perfringens* challenge in vivo and in vitro. Most importantly, AIM2-triggered inflammasome signaling activation restricts bacterial growth and proliferation. Moreover, cytosolic *C*. *perfringens*-derived genomic DNA induces AIM2-dependent inflammasome signaling activation. Collectively, these data show that AIM2 is essential for the host protection through enhancing pathogen control.

## 2. Results

### 2.1. AIM2 Contributes to Host Protection against Pathogenic C. perfringens Gas Gangrene

To assess the biological role of AIM2 in the host response to pathogenic *C*. *perfringens* infection in vivo, C57BL/6J wild-type (WT) and *Aim2*^−/−^ mice were intramuscularly challenged with 5 × 10^7^ CFUs of the log-phase pathogenic *C*. *perfringens*. The survival rate of animals was monitored for 72 h. Obviously, *Aim2*^−/−^ mice showed increased mortality rates compared with WT controls (Figure 1A). At 20 h p.i., almost 60% of *Aim2*^−/−^ mice had died, whereas all of their WT counterparts remained alive. Accordingly, *Aim2*^−/−^ mice suffered from a greater degree of severity of gas gangrene, revealed by blackening, limping, grip loss, stiffness, malaise, deformation, and swelling (Figure 1B). Next, we asked if the decreased survival and severe clinical symptoms in *Aim2*^−/−^ mice were related to increased bacterial colonization levels; mice were intramuscularly administrated with 1 × 10^7^ CFUs of the log-phase pathogenic *C*. *perfringens* to study the protective actions of AIM2 under milder conditions. As expected, dramatically increased loads of pathogenic *C*. *perfringens* were enumerated in the muscles, spleen, and liver of *Aim2*^−/−^ mice compared with their WT counterparts at 24 h p.i. (Figure 1C–E). In accordance with this, *Aim2*^−/−^ mice were subjected to more exacerbated muscle tissue injuries compared with WT controls, indicated by disrupted tissue architecture and massive inflammatory cells infiltration (Figure 1F,G). Collectively, these data suggest that AIM2 is involved in host defense against pathogenic *C*. *perfringens* gas gangrene.

### 2.2. Aim2 Deficiency Leads to Impaired Inflammasome Signaling Activation

To characterize the potential immunological mechanisms that results in increased bacterial burden in *Aim2*^−/−^ mice, we set out to evaluate the intramuscular inflammatory responses. At 24 h p.i., histological study revealed that elevated accumulation of neutrophils and macrophages in the muscles of *Aim2*^−/−^ mice compared to their WT counterparts (Figure 2A). Meanwhile, the amount of inflammatory chemokine KC was higher in the muscles of *Aim2*^−/−^ mice (Figure 2B). Crucially, IL−1β release was dramatically reduced in the muscles of *Aim2*^−/−^ mice compared to their WT counterparts, although IL−6 secretion without attaining statistical significance (Figure 2C,D). To illustrate the signaling mechanism that results in decreased IL−1β production upon log-phase pathogenic *C. perfringens* infection in the absence of *Aim2*, caspase-1 activation was firstly evaluated, a characterized protein is related to the multiprotein complex inflammasome assembly and activation. Exhilaratingly, compared to WT controls, the inflammasome-dependent caspase−1 cleavage dramatically reduced in the muscles of infected *Aim2*^−/−^ mice (Figure 2E,F), suggesting that *Aim2* deficiency impaired inflammasome signaling activation in vivo. Conclusively, these results suggested that a disturbed inflammation was associated with impaired bacterial clearance in the absence of *Aim2* during pathogenic *C. perfringens* soft tissue infection.

### 2.3. AIM2 Promotes Inflammasome Signaling Activation in Macrophages

To substantiate the pivotal role of AIM2 in muscular defense against pathogenic *C. perfringens*, AIM2 expression in the muscles of infected both genotype mice was determined by immunohistochemical staining in vivo. The results demonstrated that AIM2 staining was mainly detected in the infiltrated inflammatory cells of the infected muscles (Figure 3A). Subsequently, to describe the inflammatory responses in vitro, LPS−primed bone marrow−derived macrophages (BMDMs) derived from WT and *Aim2^−/−^* mice were infected with log-phase pathogenic *C. perfringens*. Notably, AIM2 protein expression levels were observably upregulated by bacteria challenge in WT BMDMs (Figure 3B). We next asked if *Aim2* deficiency impairs inflammasome signaling in macrophages. As shown in Figure 3C–E, inflammasome signaling was significantly attenuated in *Aim2*−deficient macrophages vs. WT macrophages, showed by impaired caspase-1 and IL−1β cleavages, and reduced mature IL−1β production, despite LPS−dependent TNF−α secretion not being suppressed in the absence of *Aim2*. Given that the upstream sensor AIM2 associates with the adaptor molecule ASC to drive assembly of inflammasome complexes and activate caspase−1, ASC speck formation is a classical event related to inflammasome activation [21]. The results revealed that *Aim2* deficiency impaired the formation of ASC specks (Figure 3F,G), indicating that pathogenic *C. perfringens* surely promoted AIM2 inflammasome signaling priming. *Aim2^−/−^* mice harbored significantly elevated loads of bacteria present in infected tissue, implying that infiltrated inflammatory cells are unable to effectively combat the multiplication of pathogenic *C. perfringens* in the absence of *Aim2*. Then, we identified the bacterial killing capacity of *Aim2*-deficient BMDMs. The results showed that *Aim2* deficiency results in a defect in the bacterial killing capacity of macrophages (Figure 3H). Thus, these observations suggested that AIM2 promotes bacterial killing and clearance.

### 2.4. AIM2-Mediated Inflammasome Signaling Facilitates Bacterial Killing and Clearance

Previous studies have implied that inflammasome signaling emerges as an important aspect of host defense [11,22]. To investigate whether AIM2-mediated inflammasome signaling activation contributes to host protection against pathogenic *C. perfringens* invasion, *Aim2^−/−^* mice were administrated with exogenous recombinant IL-18 (rIL−18) prior to pathogenic *C. perfringens* challenge, to confirm whether rIL-18 could rescue the defect in pathogenic *C. perfringens* control in the absence of *Aim2*. Strikingly, exogenous rIL−18 strongly decreased the bacterial loads in *Aim2^−/−^* mice, as shown by the viable bacteria numbers in muscle, spleen, and liver (Figure 4A–C). Indeed, infected *Aim2^−/−^* mice treated with rIL−18 exhibited remarkably alleviated pathological damage in muscles (Figure 4D). To our delight, exogenous rIL−18 could also rescue the defect of bacterial killing in *Aim2*−deficient BMDMs (Figure 4E), suggesting that AIM2 facilitates bacterial killing in an inflammasome signaling−dependent manner in macrophages. Taken together, these results indicate that the protective role of AIM2 against pathogenic *C. perfringens* infection is reliant on the capacity of AIM2 to promote inflammasome signaling activation−mediated bacterial clearance and killing.

### 2.5. Pathogenic C. perfringens-Derived Genomic DNA Mediates Activation of AIM2 Inflammasome Signaling

AIM2 is characterized as a cytosolic DNA sensor that predominantly participates in innate immunity and tumor suppression. To further elucidate how pathogenic *C. perfringens* triggers AIM2 signaling activation, LPS−primed BMDMs derived from WT and *Aim2^−/−^* mice were challenged with log−phase pathogenic *C. perfringens*−derived genomic DNA. Interestingly, *Aim2* deficiency impaired bacterial−derived genomic DNA−stimulated IL−1β release, whereas TNF−α secretion was comparable between both genotypical macrophages (Figure 5A,B), indicating that pathogenic *C. perfringens*−derived genomic DNA−triggered inflammasome signaling was attenuated in the absence of *Aim2*. In summary, our data report a novel discovery of AIM2 in the context of pathogenic *C. perfringens* infection; it recognizes bacterial−derived genomic DNA to activate inflammasome signaling, which, in turn, facilitates pathogen control and host defense.

## 3. Discussion

We unexpectedly found that AIM2 enhances bacterial killing and clearance by facilitating inflammasome signaling as a host protection against pathogenic *C*. *perfringens* infection. As a novel member of the interferon-induced PYHIN proteins, AIM2 plays a significant role in regulating innate immunity [23]. Recently, numerous studies, including ours, have shown that AIM2 is associated with pathogenic infections [24,25,26]. However, the role of AIM2 in pathogenic *C*. *perfringens* infection has not been reported. Infection with pathogenic *C*. *perfringens* can lead to necrotizing enterocolitis, small intestinal colitis, intestinal toxemia, gas gangrene, systemic diseases, and diarrhea [27,28]. These clostridial diseases not only cause incalculable economic losses to the livestock industry, but also pose a risk to human health. Meanwhile, the irrational use of antibiotics has led to the emergence of drug-resistant *C. perfringens*, which poses a series of challenges and problems. Therefore, it is urgent to develop a new therapeutic target to combat pathogenic *C. perfringens* infection. Investigating the interaction between pathogen and host is critical for effective control of *C. perfringens* infections.

Using *Aim2*^−/−^ mice, we first determined the immunological role of AIM2 in the host response to pathogenic *C*. *perfringens* soft tissue infection. We observed that *Aim2*^−/−^ mice are more susceptible to *C*. *perfringens* infection than WT controls, with lower survival, excessive inflammation, and higher bacterial colonization. This implies an essential role for AIM2 in host protection against pathogenic *C*. *perfringens* infection. Previously, *Aim2*^−/−^ mice were proved to be more susceptible to *Mycobacterium tuberculosis*, and *Aim2* deficiency resulted in impaired IL-1β production [29]. Meanwhile, it was also found that in the early stages of *Staphylococcus aureus* infection in the central nervous system, the survival rates and secretion of key inflammatory mediators were significantly decreased in *Aim2*^−/−^ mice [30]. These results suggest that AIM2-induced inflammasome signaling plays a key role in the early stages of pathogen infection. Therefore, to identify the potential signaling mechanisms underlying AIM2-mediated host defense, we set out to investigate AIM2-triggered inflammatory responses during pathogenic *C*. *perfringens* infection. We found more F4/80-labeled macrophages and Gr-1-labeled neutrophils in the muscle tissue of infected *Aim2*^−/−^ mice relative to infected WT mice. Subsequently, we examined the expression of inflammatory factors and found that *Aim2* deficiency leads to increased KC production and decreased IL-1β production. Recent studies have shown that inflammasome-activated caspase-1 cleaves inactive precursors of the IL-1 family to generate mature cytokines including IL-1β and IL-18 [10,31]. Accordingly, the inflammasome-dependent caspase-1 cleavage dramatically reduced in the muscle tissue of *Aim2*^−/−^ mice following pathogenic *C*. *perfringens* infection. Hence, these results suggest that AIM2 resistance to pathogenic *C*. *perfringens* infection may be closely linked to the inflammatory response in vivo.

Next, we further identified the potential signaling mechanism that underlies AIM2-regulated host protection. Owing to AIM2 being mainly located in the recruited inflammatory cells, macrophages were used to investigate AIM2-mediated inflammatory responses upon challenge with the pathogenic *C*. *perfringens* in vitro. Recently, increasing evidence shows that the AIM2 inflammasome can be activated in response to human papillomavirus [32], *Aspergillus fumigatus* [19], *Francisella tularensis* [24], mouse cytomegalovirus, cytosolic bacteria, and vaccinia virus [17]. However, there is little knowledge regarding the possible involvement of AIM2 in response to *C*. *perfringens*, an extracellular bacterial pathogen. We found that *Aim2* deficiency resulted in a significant inhibition of casepase-1 cleavage and IL-1β production in BMDMs, suggesting that AIM2 deficiency potentially attenuated *C*. *perfringens*-induced inflammasome signaling activation. It has been reported that the N-terminal pyrin structural domain of AIM2 triggers downstream functions via the recruitment and activation of ASC that, in turn, recruits caspase-1 and/or caspase-11 to assemble the AIM2 inflammasome [33]. Correspondingly, we showed a significant reduction in ASC speckle formation in *Aim2*^−/−^−BMDMs compared to WT-BMDMs, which further confirmed our conjecture that AIM2 resistance to *C*. *perfringens* infection may be dependent on activation of inflammasome signaling. Thus, it is worthwhile to further evaluate whether inflammasome signaling contributes to AIM2-mediated host protection. Since mature IL-18 and IL-1β are mediated by caspase-1 within inflammasome signaling [34,35], mice were subsequently pretreated with rIL-18 to evaluate if it could reverse the susceptibility of *Aim2*^−/−^ mice during pathogenic *C*. *perfringens* soft tissue infection, demonstrated by decreased bacterial loads and alleviated organ damage. Consistent with this finding, the deletion of *Aim2* obviously weakened the killing and scavenging ability of macrophages to pathogenic *C*. *perfringens*. These observations document that activation of the AIM2-mediated inflammasome signaling is indispensable in resistance to pathogenic *C. perfringens* infection.

Although AIM2 is known as a cytosolic DNA sensor that recognizes intracellular pathogenic bacteria such as *Mycobacterium tuberculosis* [36,37], *Listeria monocytogenes* [38], and *Francisella* [39], its role in sensing extracellular bacterial infection remains poorly understood. Here, we suspected that AIM2 had the potential role to recognize pathogenic *C*. *perfringens*-derived genomic DNA. Exhilaratingly, pathogenic *C*. *perfringens*-derived genomic DNA transfected into BMDMs induced mature IL-1β production. However, significantly reduced mature IL-1β release in *Aim2*^−/−^-BMDMs compared with WT-BMDMs, despite LPS-dependent TNF-α secretion being comparable, indicated that pathogenic *C*. *perfringens*-derived genomic DNA is a potential ligand for activating AIM2 inflammasome signaling. Although understanding the underlying mechanism of AIM2 inflammasome activation by pathogenic *C. perfringens* requires further investigation, these data reinforce understanding of the biological significance of AIM2 in host innate immune defense.

In conclusion, our studies document an unexpected discovery of AIM2 regulating a protective host innate immune response and preventing pathogenic *C. perfringens* gas gangrene. We provide a certain clue that inflammasome signaling downstream of AIM2 confers host defense and bacterial eradication. Also, our findings highlight the potential interplay that pathogenic *C*. *perfringens*-derived genomic DNA may serve as an agonist of AIM2 signaling. Therefore, it is promising to reveal that AIM2 is a potential therapeutic target against pathogenic invaders.

## 4. Materials and Methods

### 4.1. Mice and Cell

*Aim2*^−/−^ mice and C57BL/6J wild-type (WT) mice were gifts from Dr. Yon-Jun Yang (Jilin University, China) [20]. Mature mouse BMDMs were obtained from mouse femurs and cultured in RPMI1640 medium (Gibco, #31800-022, Waltham, MA, USA) containing 25% L929 cell-conditioned medium, 10% FBS (Gibco, #A31608-02), and 100 U/mL of penicillin/streptomycin solution (Gibco, #15140-122) for 7 days of differentiation [40].

### 4.2. Chemical

LPS was purchased from InvivoGen (#tlrl-3pelps, San Diego, CA, USA). PEI 40K Transfection Reagent was purchased from Servicebio (#G1802, Wuhan, China). Recombinant IL-18 was obtained from Novoprotein (#CK06, Suzhou, China). D-4-Amino-3-isoxazolidinone was obtained from Coolaber (#CA1662, Beijing, China). Tris-HCl (#10812846001), KCl (#1049360250), Na_3_VO_4_ (#S6508), and NaCl (#S3014) were obtained from Sigma-Aldrich (St. Louis, MO, USA).

### 4.3. In Vivo Infection

The *C. perfringens* gas gangrene model was established as previously reported [10]. Age- and sex-matched mice were inoculated intramuscularly with the log-phase *C. perfringens* strain CP1 (1 × 10^7^ or 5 × 10^7^ CFUs per mouse). *C. perfringens* gas gangrene was evaluated using previously described scoring criteria [41].

### 4.4. Bacterial Burden and Cytokine Measurements

The bacterial loads of infected mouse tissues including muscle, liver, and spleen were assessed. At 24 h p.i., aseptically excised tissues were homogenized in cold PBS and plated onto BHI (#HB8297-5, Haibo Biotechnology, Qingdao, China) agar plates for quantification of bacterial burden. For cytokine/chemokine analysis, the tissue homogenates and cell culture supernatants were detected by an ELISA assay according to the R&D systems instruction.

### 4.5. Tissue Histology and Immunofluorescence Staining

At 24 h p.i., aseptically excised tissue samples were stained with H&E (#SL7050-500, Solarbio, Beijing, China). Immunostaining was performed using anti-Ly-6G/Ly-6c (#108419, BioLegend, San Diego, CA, USA), anti-F4/80 (#123119, BioLegend), anti-AIM2 (#20590-1-AP, Proteintech, Wuhan, China), anti-ASC (#A29151803f, Adipogen, San Diego, CA, USA), Alexa Fluor 594-conjugated anti-mouse IgG (#ab150116, Invitrogen, Waltham, MA, USA) antibodies, and DAPI (Solarbio, #C0065).

### 4.6. Inflammasome Activation Assay

Mature BMDMs were pretreated with LPS (500 ng/mL, 4 h), and then, challenged with log-phase pathogenic *C. perfringens* strain CP1 or ATCC13124 (MOI = 20, 90 min), or transfected with 50 μg/mL of pathogenic *C. perfringens* strain CP1 or ATCC13124 genomic DNA for 8 h. Subsequently, samples were collected for immunoblotting, immunofluorescence staining and ELISA tests.

### 4.7. Bacterial Killing Analysis

LPS-pretreated BMDMs were pretreated with PBS or rIL-18 (1000 pg/mL, 1 h) before being challenged with log-phase pathogenic *C*. *perfringens* (MOI = 5, 6 h). The cell supernatants were plated onto BHI agar plates for bacterial quantification.

### 4.8. Administration of Exogenous rIL-18

Exogenous rIL-18 (1.0 μg per mouse) was administered intraperitoneally to *Aim2*^−/−^ mice on days 1 and 0 daily. The mice were intramuscularly inoculated with log-phase pathogenic *C*. *perfringens* CP1 (1 × 10^7^ CFUs per mouse) on day 0. Tissues were aseptically excised for bacterial load measurement and histological analysis.

### 4.9. Immunoblotting

To harvest the lysate, the infected tissues and stimulated BMDMs were lysed in a buffer containing 1% Triton X-100, 50 mM Tris-HCl, 150 mM NaCl, 0.1 mM Na_3_VO_4_, and complete protease inhibitor cocktail, and the cell supernatants were homogenized in methanol chloroform extraction. The immunoblot was performed using anti-caspase-1 (#A28881708, Adipogen, Beijing, China), anti-IL-1β (#PRP1119, Adipogen), anti-ASC, anti-AIM2, anti-GADPH (#60004-1-Ig, Proteintech, Wuhan, China), and β-Tubulin (#HC101, Transgen, Beijing, China).

### 4.10. Statistical Analysis

Date are represented as mean ± SD. Differences between mean values of normally distributed data were assessed with one-way ANOVA (Dunnett’s *t*-test) and two-tailed Student’s *t*-test. Statistical analysis for survival curves was performed using the log-rank test. Statistical significance was defined as ** p* < 0.05 and *** p* < 0.01. Statistical analysis was performed using the GraphPad Prism (version 8.0.2; La Jolla, CA, USA).

## 5. Conclusions

We focused on the biological role of AIM2 in response to pathogenic *C*. *perfringens* invading, and revealed it might elicit host antipathogenic innate immunity. The results suggest that AIM2 is required for host protection against pathogenic *C. perfringens* gas gangrene, and AIM2 mediates inflammasome signaling to promote host defense and bacterial eradication. In addition, cytosolic *C*. *perfringens*-derived genomic DNA induces AIM2-dependent inflammasome signaling activation. Therefore, this study reveals that AIM2 may be a potential therapeutic target for limiting pathogenic *C*. *perfringens* infection.

## Figures and Tables

**Figure 1 ijms-25-06571-f001:**
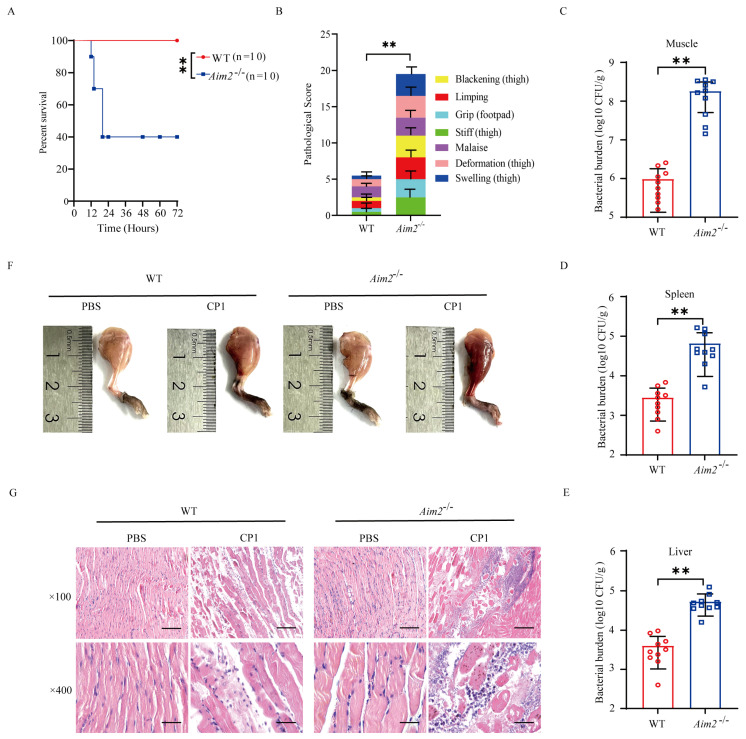
*Aim2*−deficient mice display enhanced susceptibility to pathogenic *C. perfringens* soft tissue infection. Six−to eight−week−old sex−matched C57BL/6J wild−type (WT) and *Aim2*^−/−^ mice (n = 10 per group) were intramuscularly infected with log–phase pathogenic *C*. *perfringens*. (**A**,**B**) Survival and pathologic scores (5 × 10^7^ CFUs per mouse). (**C**–**E**) Bacterial loads (1 × 10^7^ CFUs per mouse). (**F**) Representative gross images (1 × 10^7^ CFUs per mouse). (**G**) Representative hematoxylin and eosin (H&E) staining (1 × 10^7^ CFUs per mouse, magnification ×100 or ×400). Data were considered significant when *** p <* 0.01.

**Figure 2 ijms-25-06571-f002:**
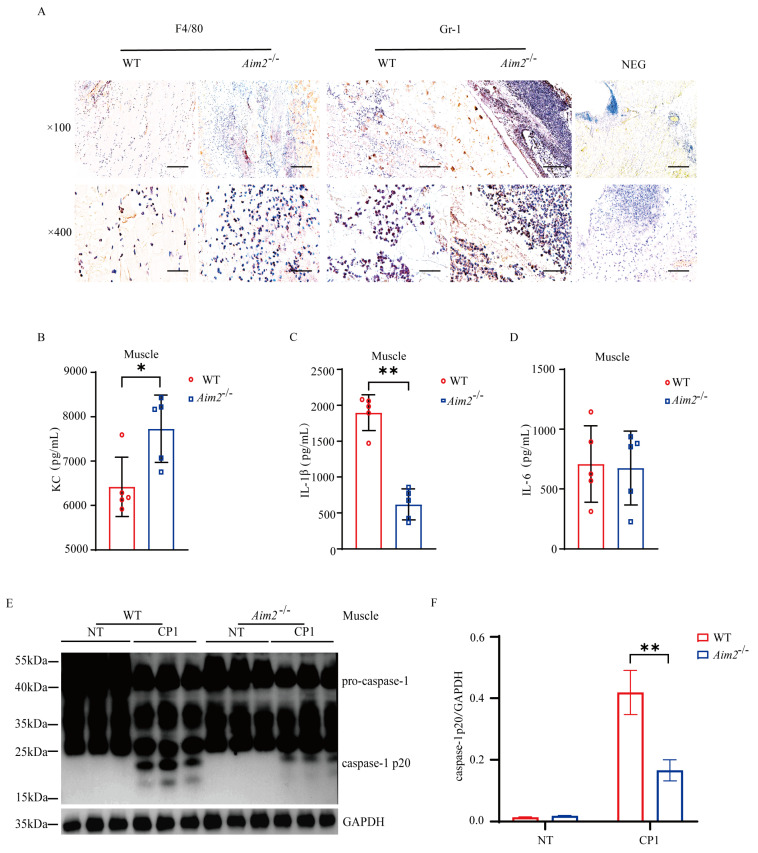
*Aim2* deficiency attenuates inflammasome signaling following pathogenic *C*. *perfringens* invasion. (**A**) Representative neutrophil and macrophageocyte (IHC) stained brown in the infected muscle sections; magnification ×100 or ×400. (**B**−**D**) The homogenate supernatant of the muscle was detected for the concentrations of KC, IL−1β, and IL−6 production by ELISA. (**E**) The muscle lysate was detected for caspase−1 expression by immunoblotting. (**F**) Amounts of caspase−1 determined by densitometry of protein bands from three experiments using imageJ software (version 1.50c, National Institutes of Health, Bethesda, MD, USA). GAPDH was used as a loading control. Data pooled five biological replicates. NEG, primary antibody omitted. Data were considered significant when * *p* < 0.05, ** *p* < 0.01.

**Figure 3 ijms-25-06571-f003:**
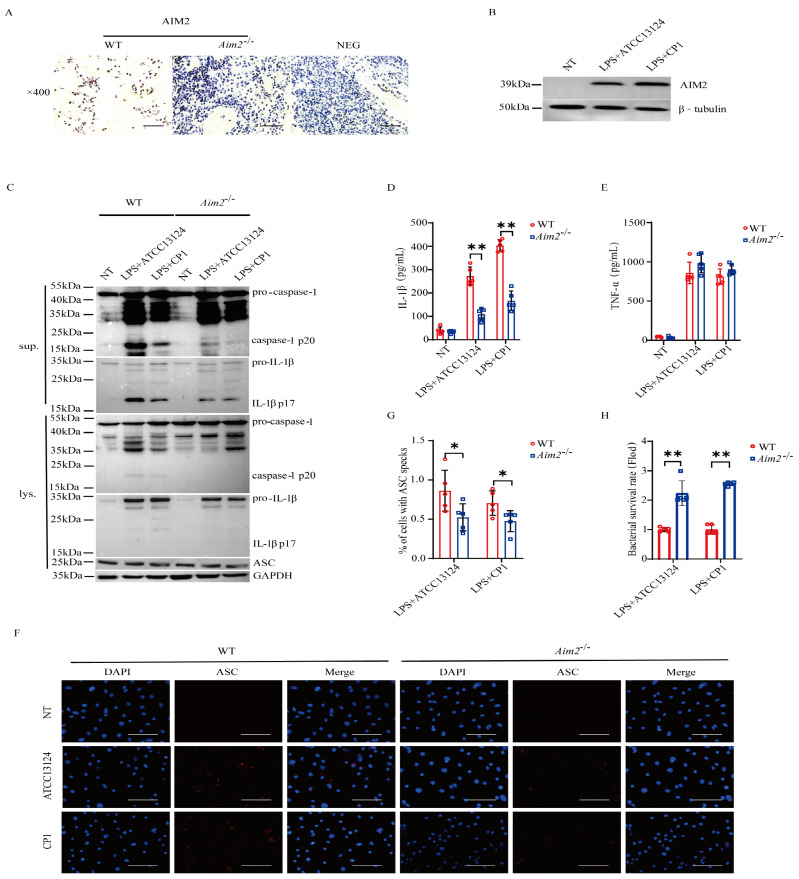
AIM2 is upregulated and triggers inflammasome signaling activation in macrophages during pathogenic *C. perfringens* challenge. (**A**) AIM2 (IHC) stained brown in the infected muscle sections, magnification ×400. LPS-primed BMDMs were challenged with log-phase pathogenic *C. perfringens* strain ATCC13124 or CP1 for 90 min (MOI = 20). (**B**,**C**) Cell supernatants and extracts immunoblotted for the expression of AIM2, caspase−1, IL−1β, ASC, β−tubulin, and GAPDH. (**D**,**E**) Culture supernatants were examined for IL−1β and TNF−α by ELISA. (**F**,**G**) ASC (IF, magnification ×400) stained red and the percentage of cells containing ASC speckles was enumerated. Nuclei stained blue. (**H**) Bacterial counts. The cells with no treatment (NT) were taken as the control group. Data pooled five biological replicates. Data were considered significant when * *p* < 0.05, ** *p* < 0.01.

**Figure 4 ijms-25-06571-f004:**
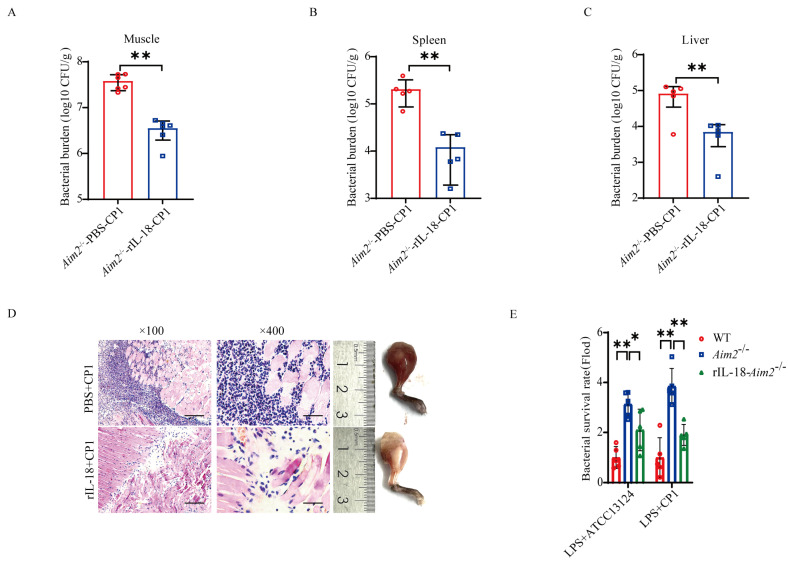
Inflammasome signaling downstream of AIM2 confers host defense against pathogenic *C. perfringens* invasion. Six−to eight−week−old sex-matched mice were administered with rIL−18 or PBS before intramuscular administration of log-phase pathogenic *C. perfringens* (n = 5 each group). (**A**−**C**) Bacterial counts. (**D**) Representative gross images and H&E staining of infected muscle (magnification ×100 or ×400). LPS−pretreated BMDMs were incubated with PBS or rIL−18 before the infection of log−phase pathogenic *C. perfringens*. (**E**) Bacterial loads. Data pooled five biological replicates. Data were considered significant when * *p* < 0.05, ** *p* < 0.01.

**Figure 5 ijms-25-06571-f005:**
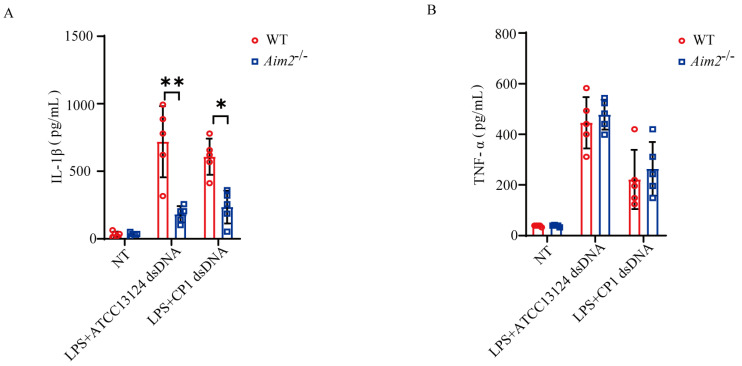
Inflammasome signaling activation in response to cytosolic C. perfringens−derived genomic DNA is dependent on AIM2. LPS−pretreated BMDMs were transfected with 50 μg/mL of pathogenic C. perfringens genomic DNA using PEI 40K. (**A**,**B**) Culture supernatants were examined for IL−1β and TNF−α production by ELISA. Data pooled five biological replicates. Data were considered significant when * *p* < 0.05, ** *p* < 0.01.

## Data Availability

The data that support the findings of this study are available from the corresponding author upon reasonable request.

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
