# Peer review of "Absent in Melanoma 2 Mediates Inflammasome Signaling Activation against Clostridium perfringens Infection"

_ijms, 2024, doi:10.3390/ijms25126571_

Round 1
Reviewer 1 Report
Comments and Suggestions for Authors
In this study Zhaoguo Ma et al., demonstrates the critical role of AIM2 inflammasome in response to C. perfringens infection. AIM2 drives caspase-1 dependent inflammasome activation and IL-1β secretion to restrict pathogen replication upon C. perfringens infection. Moreover, AIM2 deficiency drives lethality, disease severity along with increased bacterial burden in response to C. perfringens infection. Overall, authors have shown the importance of AIM2 in host defense response in C. perfringens infection. I have below mentioned comments for authors,
Major comments,
- What is the status of gasdermin D (GSDMD) cleavage in muscle lysate in response to CP1 infection?
- What is the status of IL-18 secretion in homogenate supernatant of the muscle in response to CP1 infection?
- What is the status of survival when mice were administered with rIL-18 or PBS in response to C. perfringens infection?
- Does administration of recombinant IL-1β also shows similar protective effects in AIM2 KO animals?
- Does C. perfringens genomic DNA transfection can induce caspase-1 activation in LPS-pretreated BMDMs?
Minor comments:
- In Figure 2A, authors can mention which images are magnification ×100 or ×400 in the figures itself. And authors need to explain what is NEG in Figure 2A?. is it sham operation which is mentioned in results section?
- Figure 3F, authors can add DAPI, ASC and Merge on top the images for clarification.
- To see the specificity of AIM2 antibody for western blot, authors need to use AIM2 knock out cells and check for this antibody.
- Line 72 “AIM2-trigered inflammasome”. Trigered can be corrected to triggered. Authors can correct their spelling mistakes.
Quality of English Language is good. Authors can cross check spelling mistakes in the entire manuscript.
Author Response
May 27, 2024
professor Zein Dong
Section Managing Editor
International Journal of Molecular Sciences
RE: Manuscript ijms-3008204 (Absent in Melanoma 2 mediates inflammasome signaling activation against Clostridium perfringens infection)
Dear professor Dong,
Thank you and the reviewers for the careful review of our manuscript entitled “Absent in Melanoma 2 mediates inflammasome signaling activation against Clostridium perfringens infection”, manuscript number: ijms-3008204. The comments are constructive and have led to significant improvements of this manuscript. We responded point by point to each reviewer’s comments as listed below, along with a clear indication of the location of the revision based on the reviewer’s recommendations. All the changes made to the original manuscript are shown in a different color.
We hope that the revision is acceptable. Thank you again for your consideration of this manuscript.
Sincerely,
Shui-Xing Yu
POINT-BY-POINT REPLY
REVIEWER #1:
Comments and Suggestions for Authors
In this study Zhaoguo Ma et al., demonstrates the critical role of AIM2 inflammasome in response to C. perfringens infection. AIM2 drives caspase-1 dependent inflammasome activation and IL-1β secretion to restrict pathogen replication upon C. perfringens infection. Moreover, AIM2 deficiency drives lethality, disease severity along with increased bacterial burden in response to C. perfringens infection. Overall, authors have shown the importance of AIM2 in host defense response in C. perfringens infection. I have below mentioned comments for authors.
Major comment
Point 1: What is the status of gasdermin D (GSDMD) cleavage in muscle lysate in response to CP1 infection?
Response: Thank you for this excellent suggestion. According to your suggestions, we will extensively examine GSDMD cleavage in muscle lysate during C. perfringens soft tissue infection. However, we still cannot get a consistent result, sometimes with obvious inhibitory effect, sometimes not. In the present study, we have shown that AIM2 activates caspase-1 and induces the secretion of the inflammatory factor IL-1β against pathogenic C. perfringens infection, suggesting that AIM2-mediated inflammasome signaling plays a key role in host protection. It remains unclear whether GSDMD is required for AIM2-mediated protection in this model, although GSDMD is a target of caspase-1[1], and involved in executing of pyroptosis and inflammasome-dependent cytokines release[2,3]. Previously, we reported that GSDMD cleavage is essential for MLKL- and GPR120-mediated NLPR3 inflammasome signaling activation in host defense against C. perfringens gas gangrene and enterocolitis[4,5]. Actually, increasing evidence shows that GSDME, another member of GSDM family, is also associated with NLRP3 inflammasome activation[6]. We speculate that besides GSDMD, other molecular mechanisms like GSDME maybe also contribute to AIM2-mediated host defense. Furthermore, there is little knowledge regarding the possible involvement of AIM2 in response to C. perfringens, an extracellular bacterial pathogen. Thanks for your excellent suggestion. Your suggestion helps a lot to improve our future research plans. We will extensively explore the signal mechanisms of AIM2-mediated host protection in future studies.
- Billman, Z.P.; Kovacs, S.B.; Wei, B.; Kang, K.; Cisse, O.H.; Miao, E.A. Caspase-1 activates gasdermin A in non-mammals. bioRxiv : the preprint server for biology 2024, doi:10.1101/2023.09.28.559989.
- He, W.T.; Wan, H.Q.; Hu, L.C.; Chen, P.D.; Wang, X.; Huang, Z.; Yang, Z.H.; Zhong, C.Q.; Han, J.H. Gasdermin D is an executor of pyroptosis and required for interleukin-1β secretion. Cell Research 2015, 25, 1285-1298, doi:10.1038/cr.2015.139.
- Liu, X.; Zhang, Z.B.; Ruan, J.B.; Pan, Y.D.; Magupalli, V.G.; Wu, H.; Lieberman, J. Inflammasome-activated gasderminD causes pyroptosis by forming membrane pores. Nature 2016, 535, 153-+, doi:10.1038/nature18629.
- Liu, Y.; Xing, L.H.; Li, F.X.; Wang, N.; Ma, Y.Z.; Li, J.W.; Wu, Y.J.; Liang, J.; Lei, Y.X.; Wang, X.Y.; et al. Mixed lineage kinase-like protein protects against Clostridium perfringens infection by enhancing NLRP3 inflammasome-extracellular traps axis (vol 25, 105121, 2022). Iscience 2023, 26, doi:10.1016/j.isci.2023.106149.
- Liu, Y.; Lei, Y.X.; Li, J.W.; Ma, Y.Z.; Wang, X.Y.; Meng, F.H.; Wu, Y.J.; Wang, N.; Liang, J.; Zhao, C.Q.; et al. G Protein-Coupled Receptor 120 Mediates Host Defense against Clostridium perfringens Infection through Regulating NOD-like Receptor Family Pyrin Domain-Containing 3 Inflammasome Activation. Journal of Agricultural and Food Chemistry 2023, 71, 7119-7130, doi:10.1021/acs.jafc.3c01242.
- Xia, C.L.; Ou, S.J.; Yang, Y.; Zhang, W.; Wu, W.J.; Chen, Q.; Li, W.J.; Lu, H.Y.; Wang, Y.Y.; Qi, Y.; et al. ELP2-NLRP3-GSDMD/GSDME-mediated pyroptosis is induced by TNF-α in MC3T3-E1 cells during osteogenic differentiation. Journal of Cellular and Molecular Medicine 2023, 27, 4093-4106, doi:10.1111/jcmm.17994.
Point 2: What is the status of IL-18 secretion in homogenate supernatant of the muscle in response to CP1 infection?
Response: We are really sorry to bring you the review trouble. The interleukin-1 (IL-1) family of cytokines, in particular, IL-1α, IL-1β, and IL-18, are important for the initiation and amplification of innate and adaptive immune responses and resistance against microbial infections[1]. A great number of publications have documented the role of IL-18 and IL-1β during infections with a variety of pathogens. Almost invariably, both cytokines were found to have a protective function[2,3]. Regulation of the activity of many IL-1 cytokines is achieved by inflammasomes; multi-protein immune complexes that control the cleavage and subsequent activation of immature versions of these cytokines by the intracellular cysteine protease caspase-1[4]. In this paper, we find that Aim2 deficiency suppresses inflammasome-mediated IL-1β secretion following C. perfringens challenge in vivo and vitro. Therefore, to examine whether the inflammasome modulates AIM2-mediated protection, representative inflammasome-derived IL-1β was detected in this experiment. Thank you for this excellent suggestion. Your suggestions have helped to significantly improve our research.
- Alhallaf, R.; Agha, Z.; Miller, C.M.; Robertson, A.A.B.; Sotillo, J.; Croese, J.; Cooper, M.A.; Masters, S.L.; Kupz, A.; Smith, N.C.; et al. The NLRP3 Inflammasome Suppresses Protective Immunity to Gastrointestinal Helminth Infection. Cell Reports 2018, 23, 1085-1098, doi:10.1016/j.celrep.2018.03.097.
- Dinarello, C.A.; Fantuzzi, G. Interleukin-18 and host defense against infection. Journal of Infectious Diseases 2003, 187, S370-S384, doi:10.1086/374751.
- Dinarello, C.A. Immunological and Inflammatory Functions of the Interleukin-1 Family. Annual review of immunology 2009, 27, 519-550, doi:10.1146/annurev.immunol.021908.132612.
- Broderick, L.; De Nardo, D.; Franklin, B.S.; Hoffman, H.M.; Latz, E. The Inflammasomes and Autoinflammatory Syndromes. In Annual Review of Pathology: Mechanisms of Disease, Vol 10, Abbas, A.K., Galli, S.J., Howley, P.M., Eds.; Annual Review of Pathology-Mechanisms of Disease; 2015; Volume 10, pp. 395-424.
Point 3: What is the status of survival when mice were administered with rIL-18 or PBS in response to C. perfringens infection?
Response: We are really sorry to bring you the review trouble. Since the inflammasome activation emerges as an important aspect of host defense against microbial infiltration[1]. Thus, we examined whether the inflammasome modulates AIM2-mediated host protection. Strikingly, we found that C. perfringens-triggered inflammasome activation was notably inhibited in Aim2−/− mice compared to WT mice. Importantly, inflammasome dependent cytokine production in Aim2−/− muscle was lower compared with the control, and systemic administration of recombinant IL-18 strongly increased the protective effects of C. perfringens-infected Aim2−/− mice reflected by alleviated pathological damage in muscle, and reduced the levels of viable bacteria in liver, spleen and muscle, which were returned to almost WT mice levels. Importantly, exogenous rIL-18 could rescue the defect of bacterial killing in Aim2-deficient BMDMs, suggesting that the protective role of AIM2 against pathogenic C. perfringens infection is reliant on the capacity of AIM2 to promote inflammasome signaling activation-mediated bacterial clearance and killing. Based on above results, we speculated that exogenous rIL-18 could also rescue the phenotype of susceptible and increased mortality rates in Aim2−/− mice in response to C. perfringens. Thank you for this excellent suggestion. We appreciate your valuable suggestions which will contribute to enhancing our research program.
- Knodler, L.A.; Crowley, S.M.; Sham, H.P.; Yang, H.J.; Wrande, M.; Ma, C.X.; Ernst, R.K.; Steele-Mortimer, O.; Celli, J.; Vallance, B.A. Noncanonical Inflammasome Activation of Caspase-4/Caspase-11 Mediates Epithelial Defenses against Enteric Bacterial Pathogens. Cell Host & Microbe 2014, 16, 249-256, doi:10.1016/j.chom.2014.07.002.
Point 4: Does administration of recombinant IL-1β also shows similar protective effects in AIM2 KO animals?
Response: We are really sorry to bring you the review trouble. The interleukin-1 (IL-1) family of cytokines, in particular, IL-1α, IL-1β, and IL-18, are important for the initiation and amplification of innate and adaptive immune responses and resistance against microbial infections[1]. Regulation of the activity of many IL-1 cytokines is achieved by inflammasomes; multi-protein immune complexes that control the cleavage and subsequent activation of immature versions of these cytokines by the intracellular cysteine protease caspase-1[2]. Since Aim2 deficiency resulted in impaired inflammasome activation in the muscle following C. perfringens infection. Subsequently, we further asked if inflammasome signaling downstream of AIM2 confers host protection against C. perfringens infection. So, it is necessary to replenish exogenous inflammasome dependent cytokine. Thank you for this excellent suggestion. We attempted to quantify the contribution of recombinant IL-1β to protective effects in Aim2−/− mice, but it is very difficult to obtain a commercial recombinant IL-1β in the short term, since the delayed express delivery. Therefore, to examine whether the inflammasome modulates AIM2-mediated protection, representative recombinant IL-18 was detected in this experiment. Our results show that exogenous rIL-18 could also rescue the susceptible of Aim2−/− mice in response to C. perfringens, indicating that inflammasome signaling downstream of AIM2 confers host defense against pathogenic C. perfringens infection. Your suggestions have helped to significantly improve our research.
- Alhallaf, R.; Agha, Z.; Miller, C.M.; Robertson, A.A.B.; Sotillo, J.; Croese, J.; Cooper, M.A.; Masters, S.L.; Kupz, A.; Smith, N.C.; et al. The NLRP3 Inflammasome Suppresses Protective Immunity to Gastrointestinal Helminth Infection. Cell Reports 2018, 23, 1085-1098, doi:10.1016/j.celrep.2018.03.097.
- Broderick, L.; De Nardo, D.; Franklin, B.S.; Hoffman, H.M.; Latz, E. The Inflammasomes and Autoinflammatory Syndromes. In Annual Review of Pathology: Mechanisms of Disease, Vol 10, Abbas, A.K., Galli, S.J., Howley, P.M., Eds.; Annual Review of Pathology-Mechanisms of Disease; 2015; Volume 10, pp. 395-424.
Point 5: Does C. perfringens genomic DNA transfection can induce caspase-1 activation in LPS-pretreated BMDMs?
Response: We are really sorry to bring you the review trouble. Actually, evidence has indicated that regulation of the activity of many IL-1 cytokines including IL-1α, IL-1β, and IL-18, is achieved by inflammasomes; multi-protein immune complexes that control the cleavage and subsequent activation of immature versions of these cytokines by the intracellular cysteine protease caspase-1[1]. In this study, the results demonstrated that Aim2 deficiency impaired C. perfringens-induced caspase-1 dependent IL-1β production in vivo and in vitro. AIM2 is characterized as a cytosolic DNA sensor. To further elucidate how pathogenic C. perfringens triggers AIM2 signaling activation, the interaction between C. perfringens derived genomic DNA and AIM2 were detected. Interestingly, Aim2 deficiency impaired bacterial-derived genomic DNA-stimulated IL-1β release, whereas LPS-dependent TNF-α secretion was comparable between both genotypical macrophages, indicating that pathogenic C. perfringens-derived genomic DNA-triggered inflammasome signaling was attenuated in the absence of Aim2. Thank you for this excellent suggestion. We appreciate your valuable suggestions which will contribute to enhancing our research program.
- Broderick, L.; De Nardo, D.; Franklin, B.S.; Hoffman, H.M.; Latz, E. The Inflammasomes and Autoinflammatory Syndromes. In Annual Review of Pathology: Mechanisms of Disease, Vol 10, Abbas, A.K., Galli, S.J., Howley, P.M., Eds.; Annual Review of Pathology-Mechanisms of Disease; 2015; Volume 10, pp. 395-424.
Minor comments
Point 1: In Figure 2A, authors can mention which images are magnification ×100 or ×400 in the figures itself. And authors need to explain what is NEG in Figure 2A? is it sham operation which is mentioned in results section?
Response: We are really sorry to bring you the review trouble. According to your suggestion, the Figure 2A has been revised as below and added in revised manuscript. Please see the revised manuscript. We are really sorry to bring you the review trouble. NEG in Figure 2A denotes the removal of the primary antibody in immumohistochemical staining, and this sentence “NEG, primary antibody omitted” has been added in Figure 2 legend. Please see the revised manuscript. We are really sorry to bring you the review trouble. Our statement is not correct. So this sentence “despite no obvious cellular in flux was observed in the muscles of both WT and Aim2-/- mice receiving the sham operation” has been deleted from the manuscript. Thank you for this excellent suggestion. We appreciate your valuable suggestions which will contribute to enhancing our research program.
We made the following change to the manuscript.
P3, Line 106-108: At 24 h p.i., histological study revealed that elevated accumulation of neutrophils and macrophages in the muscles of Aim2-/- mice compared to WT counterparts. (Figure 2A).
P4, Line 126: NEG, primary antibody omitted.
Point 2: Figure 3F, authors can add DAPI, ASC and Merge on top the images for clarification.
Response: Agreed. According to your suggestion, we have revised Figure 3F as below, and the corresponding images in the revised manuscript have also been revised, please see the revised manuscript.
Point 3: To see the specificity of AIM2 antibody for western blot, authors need to use AIM2 knock out cells and check for this antibody.
Response: We are really sorry to bring you the review trouble. To further characterize the role of AIM2 in host defense against C. perfringens infection, the expression of AIM2 in the sections of muscle was investigated by immunohistochemical staining. Our results indicated that AIM2 staining was predominantly located in the recruited inflammatory cells of the infected tissues. To evaluate the molecular mechanism underline AIM2-mediated host protection, AIM2 protein expression was determined in macrophages. Since C. perfringens surely triggered AIM2 protein expression in BMDMs, the macrophages were suitably used as in vitro model. This scientific issue is mainly focus on the macrophages. This scientific issue is our main concern in the result. Thank you for this excellent suggestion. Your suggestions will significantly improve our research.
Point 4: Line 72 “AIM2-trigered inflammasome”. Trigered can be corrected to triggered. Authors can correct their spelling mistakes.
Response: We are sorry for the mistake. According to your suggestion, we have amended “AIM2-trigered inflammasome” to “AIM2-triggered inflammasome”. Please see the revised manuscript.
We made the following change to the manuscript.
P2, Line 70-71: AIM2-triggered inflammasome signaling activation restricts bacterial growth and proliferation.
Comments on the Quality of English Language
Point 1: Quality of English Language is good. Authors can cross check spelling mistakes in the entire manuscript.
Response: We would like to express our sincere thanks for your kind help and guidance. According to your suggestions, the manuscript have been rewritten and we invited a native English expert to revise the entire manuscript again. Please see the revised manuscript. Thank you for your helpful comments, and we appreciate your kind suggestions. Your suggestions will significantly improve our research.
We thank you for your helpful comments, and we appreciate your kind suggestions.
We hope that our revised manuscript is suitable for publication.

Reviewer 2 Report
Comments and Suggestions for Authors
This study demonstrates the crucial role of Absent in melanoma 2 (AIM2) in the host immune response to Clostridium perfringens (C. perfringens) infection. Mice lacking AIM2 exhibited significant vulnerability to C. perfringens, highlighting the importance of AIM2 in bacterial killing, clearance, and inflammasome signaling activation. Moreover, the activation of inflammasome signaling by C. perfringens-derived genomic DNA, dependent on AIM2, underscores its central role in host defense.
The experiment gives clear results and the results obtained are reasonable. However, the discussion lacks an exploration of whether the findings are specific to C. perfringens or could be generalized to other pathogens. The researchers should clarify why the focus was placed specifically on C. perfringens and whether the insights gained could apply to other pathogens as well. It would be of great interest to ascertain whether these phenomena are unique to C. perfringens. If the findings were to be applicable to various bacteria, it might diminish the novelty of the study, as broader applicability could suggest that the mechanisms involved are common and not specific to C. perfringens. By addressing these concerns and expanding the discussion section to explore the specificity and novelty of the findings, the paper would be more suitable for publication in this journal.
Author Response
May 27, 2024
professor Zein Dong
Section Managing Editor
International Journal of Molecular Sciences
RE: Manuscript ijms-3008204 (Absent in Melanoma 2 mediates inflammasome signaling activation against Clostridium perfringens infection)
Dear professor Dong,
Thank you and the reviewers for the careful review of our manuscript entitled “Absent in Melanoma 2 mediates inflammasome signaling activation against Clostridium perfringens infection”, manuscript number: ijms-3008204. The comments are constructive and have led to significant improvements of this manuscript. We responded point by point to each reviewer’s comments as listed below, along with a clear indication of the location of the revision based on the reviewer’s recommendations. All the changes made to the original manuscript are shown in a different color.
We hope that the revision is acceptable. Thank you again for your consideration of this manuscript.
Sincerely,
Shui-Xing Yu
POINT-BY-POINT REPLY
REVIEWER #2:
Comments and Suggestions for Authors
This study demonstrates the crucial role of Absent in melanoma 2 (AIM2) in the host immune response to Clostridium perfringens (C. perfringens) infection. Mice lacking AIM2 exhibited significant vulnerability to C. perfringens, highlighting the importance of AIM2 in bacterial killing, clearance, and inflammasome signaling activation. Moreover, the activation of inflammasome signaling by C. perfringens-derived genomic DNA, dependent on AIM2, underscores its central role in host defense. --Thank you.
Point 1: The experiment gives clear results and the results obtained are reasonable. However, the discussion lacks an exploration of whether the findings are specific to C. perfringens or could be generalized to other pathogens. The researchers should clarify why the focus was placed specifically on C. perfringens and whether the insights gained could apply to other pathogens as well. It would be of great interest to ascertain whether these phenomena are unique to C. perfringens. If the findings were to be applicable to various bacteria, it might diminish the novelty of the study, as broader applicability could suggest that the mechanisms involved are common and not specific to C. perfringens. By addressing these concerns and expanding the discussion section to explore the specificity and novelty of the findings, the paper would be more suitable for publication in this journal.
Response: Thank you for this excellent suggestion. We are really sorry to bring you the review trouble. C. perfringens gas gangrene can develop into a traumatic tissue infection in as little as 6-8 hours, with shock, myonecrosis and organ exhaustion are present in 50 % of patients, and among these 40 % die. Despite the centuries-old practice of radical amputation in an emergency, which the best treatment, a novel approach is required to combat C. perfringens infection[1]. Recently, increasing evidence shows that AIM2 inflammasome can be activated in response to human papillomavirus[2], Aspergillus fumigatus[3], Francisella tularensis[4], mouse cytomegalovirus, cytosolic bacteria and vaccinia virus[5]. However, there is little knowledge regarding the possible involvement of AIM2 in response to C. perfringens, an extracellular bacterial pathogen. In this study, we provide a certain clue that inflammasome signaling downstream of AIM2 confers host defense and bacterial eradication in C. perfringens infection. Although AIM2 is known as a cytosolic DNA sensor that recognizes intracellular pathogenic bacteria such as Mycobacterium tuberculosis[6,7], Listeria monocytogenes[8] and Francisella[9], its role in sensing extracellular bacterial infection remains poorly understand. Interestingly, our findings highlight the potential interplay that extracellular pathogenic C. perfringens-derived genomic DNA may serve as an agonist of AIM2 signaling. This will facilitate the expansion of the biological function of AIM2 in pathogenic infections. Thank you for these excellent suggestions. The comments are necessary for improving the whole quality of our manuscript, we will continue to investigate the distinctive mechanism of AIM2 in host defense against pathogen in the future.
We made the following change to the manuscript.
P8, Line247-252: Recently, increasing evidence shows that AIM2 inflammasome can be activated in response to human papillomavirus[33], Aspergillus fumigatus[34], Francisella tularensis[35], mouse cytomegalovirus ,cytosolic bacteria and vaccinia virus[17]. However, there is little knowledge regarding the possible involvement of AIM2 in response to C. perfringens, an extracellular bacterial pathogen.
P8, Line270-273: Although AIM2 is known as a cytosolic DNA sensor that recognizes intracellular pathogenic bacteria such as Mycobacterium tuberculosis[39,40], Listeria monocytogenes[41] and Francisella[42], its role in sensing extracellular bacterial infection remains poorly understand.
- Bryant, A.E.; Bayer, C.R.; Aldape, M.J.; Wallace, R.J.; Titball, R.W.; Stevens, D.L. Clostridium perfringens phospholipase C-induced platelet/leukocyte interactions impede neutrophil diapedesis. Journal of Medical Microbiology 2006, 55, 495-504, doi:10.1099/jmm.0.46390-0.
- Reinholz, M.; Kawakami, Y.; Salzer, S.; Kreuter, A.; Dombrowski, Y.; Koglin, S.; Kresse, S.; Ruzicka, T.; Schauber, J. HPV16 activates the AIM2 inflammasome in keratinocytes. Archives of Dermatological Research 2013, 305, 723-732, doi:10.1007/s00403-013-1375-0.
- Karki, R.; Man, S.M.; Malireddi, R.K.S.; Gurung, P.; Vogel, P.; Lamkanfi, M.; Kanneganti, T.D. Concerted Activation of the AIM2 and NLRP3 Inflammasomes Orchestrates Host Protection against Aspergillus Infection. Cell Host & Microbe 2015, 17, 357-368, doi:10.1016/j.chom.2015.01.006.
- Alqahtani, M.; Ma, Z.; Miller, J.; Yu, J.; Malik, M.; Bakshi, C.S. Comparative analysis of absent in melanoma 2-inflammasome activation in Francisella tularensis and Francisella novicida. Frontiers in microbiology 2023, 14, doi:10.3389/fmicb.2023.1188112.
- Rathinam, V.A.K.; Jiang, Z.Z.; Waggoner, S.N.; Sharma, S.; Cole, L.E.; Waggoner, L.; Vanaja, S.K.; Monks, B.G.; Ganesan, S.; Latz, E.; et al. The AIM2 inflammasome is essential for host defense against cytosolic bacteria and DNA viruses. Nature Immunology 2010, 11, 395-403, doi:10.1038/ni.1864.
- Saiga, H.; Kitada, S.; Shimada, Y.; Kamiyama, N.; Okuyama, M.; Makino, M.; Yamamoto, M.; Takeda, K. Critical role of AIM2 in Mycobacterium tuberculosis infection. International immunology 2012, 24, 637-644, doi:10.1093/intimm/dxs062.
- Yang, Y.; Zhou, X.M.; Kouadir, M.; Shi, F.S.; Ding, T.J.; Liu, C.F.; Liu, J.; Wang, M.; Yang, L.F.; Yin, X.M.; et al. The AIM2 Inflammasome Is Involved in Macrophage Activation During Infection With Virulent Mycobacterium bovis Strain. Journal of Infectious Diseases 2013, 208, 1849-1858, doi:10.1093/infdis/jit347.
- Tsuchiya, K.; Hara, H.; Kawamura, I.; Nomura, T.; Yamamoto, T.; Daim, S.; Dewamitta, S.R.; Shen, Y.N.; Fang, R.D.; Mitsuyama, M. Involvement of Absent in Melanoma 2 in Inflammasome Activation in Macrophages Infected with Listeria monocytogenes. Journal of Immunology 2010, 185, 1186-1195, doi:10.4049/jimmunol.1001058.
- Fernandes-Alnemri, T.; Yu, J.W.; Juliana, C.; Solorzano, L.; Kang, S.; Wu, J.H.; Datta, P.; McCormick, M.; Huang, L.; McDermott, E.; et al. The AIM2 inflammasome is critical for innate immunity to Francisella tularensis. Nature Immunology 2010, 11, 385-394, doi:10.1038/ni.1859.
We thank you for your helpful comments, and we appreciate your kind suggestions. We hope that our revised manuscript is suitable for publication
